# Yield Estimation of Wheat Using Cropland Masks from European Common Agrarian Policy: Comparing the Performance of EVI2, NDVI, and MTCI in Spanish NUTS-2 Regions

M. A. Garcia-Perez *, V. Rodriguez-Galiano [ID], E. Sanchez-Rodriguez [ID] and V. Egea-Cobrero

Department of Physical Geography and Regional Geographic Analysis, Universidad de Sevilla,
C/Doña María de Padilla, S/N, 41004 Sevilla, Spain; vrgaliano@us.es (V.R.-G.); esanchez@us.es (E.S.-R.)
* Correspondence: mgperez@us.es; Tel.: +34-600644728

**Abstract:** Monitoring wheat yield and production is essential for ensuring global food security. Remote sensing can be used to achieve it due to its ability to provide global, comprehensive, synoptic, and repetitive information in near real-time. This study used the 2006–2016 Normalized Difference Vegetation Index (NVDI) and Enhanced Vegetation Index 2 (EVI2) time series at a 250 m spatial resolution and 2006–2011 MERIS Terrestrial Chlorophyll Index (MTCI) time series at a 300 m spatial resolution. The post-maximum period for pixels containing wheat was selected based on the EU's Common Agrarian Policy (CAP) and Corine Land Cover (CLC) data. It was correlated with yield and production values from governmental statistics (GS) of the largest Nomenclature of Territorial Units for Statistics level 2 (NUTS-2) wheat producers in Spain and for Spain overall. The selection of wheat masks was crucial for the accuracy of the models, with CAP masks offering greater forecasting capability. Models using CLC produced $R^2$ values between 0.45 and 0.7, while those using CAP outperformed the former with $R^2$ values of 0.9 throughout Spain. Production models outperformed yield models, and MTCI was the vegetation index (VI) that provided the greatest $R^2$ value of 0.94. However, model accuracy was heavily conditioned by the precision of input data, where anomalies were detected in some NUTS-2.

**Keywords:** Common Agrarian Policy; crop yield; crop production; vegetation index

## 1. Introduction

Wheat harvest is essential for ensuring global food security [1], which is understood as the right to access adequate nutrition in order to sustain an active and healthy life [2]. Wheat is the most cultivated crop in the world after maize, serving as the nutritional basis for the majority of the global population and accounting for 20% of the daily caloric intake of 4.5 billion people [3]. Wheat is cultivated around the world, although it is especially concentrated in Europe, eastern China, northern India, and the interior of the United States. Wheat croplands have increased by 5.72% between 1961 and 2019 [4], and, thanks to technological advances, production has increased at an even faster rate, from an average production of 1089 kg ha$^{-1}$ in 1961 to 3546.8 kg ha$^{-1}$ in 2019 [4]. World wheat production has therefore increased from 222,357,231 tons in 1961 to 765,769,635 tons in 2019. In 2019, the primary producer of wheat was China, at 133,596,300 tons, followed by India and the United States [4]. At the same time, the global population also increased, from 3.073 billion in 1961 to 7.673 billion in 2019. This means there was a 249.69% population increase in the past 58 years [5]. It is expected that, by 2050, the world's population will exceed 9 billion people, and wheat demand will increase by 60% compared to 2011 [3]. Moreover, global climate change can have a direct impact on crop growth and production [6]. Temperature increase has had a negative impact on wheat production in India [1]. Climate change between 1980 and 2008 may have reduced potential global wheat production by 5.5% [7]. In [8], they concluded that global wheat production will decrease by 6% for each degree

Celsius increase in mean global temperature. Moreover, an increase in the occurrence and intensity of extreme weather events is expected as a result of the effects of climate change [9]. Thus, agriculture must be more efficient, intensifying crops in a sustainable manner and avoiding environmental degradation [10].

Understanding wheat production and yield is important for designing national and international agricultural policies and ensuring food supply and security [11,12]. There are several international initiatives to monitor crops and estimate their production [13]: Global Agricultural Monitoring (GeoGLAM) [14], Global Information and Early Warning System (GIEWS) [15], AquaCrop [16], and Monitoring Agricultural Resources (MARS) [17], among others. Agronomic models are used to estimate and forecast crop production [6]. On one hand, the use of crop forecast models enables us to understand crop production prior to the harvesting period. They are based on several near real-time variables (climate, ground properties, crop cultivation techniques, remote sensing-derived variables, etc.). Their aim is to minimize risks and to modify crop management when a problem is detected in order to obtain optimal crop production. According to [18], several techniques have been employed to obtain the input data for these models, such as in situ crop observation, survey sampling, the previous year's production, or the simulation of crop growth—methodologies described in [19–21]. On the other hand, harvest estimation models aim to describe the productive potential of agricultural regions in order to control their evolution over time [22–27]. This type of model compares different biotic and abiotic parameters (e.g., health status of the plant, photosynthetic activity, and water availability) to observed crop production, thereby highlighting possible disparities between food production and demand on a regional scale [28]. In [29], they classified models as knowledge-driven and data-driven. Knowledge-driven models are based on a theoretical understanding of crop growth and development, as well as knowledge of the primary physiological mechanisms of vegetative cover [30]. Data-driven models are based on an inductive focus that compares crop yield to a series of variables and do not take into account any explicit theoretical knowledge that may impact vegetative behavior. Several variables that may affect production are selected (e.g., radiation intercepted by vegetation, temperature, humidity, etc.), and regression models are applied to determine whether or not a correlation exists between these variables and crop production [31,32].

Remote sensing data are widely used as an independent variable for data-driven models. It provides information on the photosynthetic activity of vegetative cover, such as the Fraction of Absorbed Photosynthetically Active Radiation (FAPAR) [33] and the Leaf Area Index (LAI) [34], two aspects that are highly correlated with crop yield [35,36]. Some sensors used for studies on a regional or national scale are the Advanced Very High Resolution Radiometer (AVHRR), the Moderate-Resolution Imaging Spectroradiometer (MODIS) sensor, and the Medium-Resolution Imaging Spectrometer (MERIS) sensor. They can be used to calculate several VIs, e.g., Normalized Difference Vegetation Index (NDVI), Enhanced Vegetation Index 2 (EVI2), and MERIS Terrestrial Chlorophyll Index (MTCI). VIs have been used in several studies that employ wheat estimation and forecasting models on various scales, from local studies that focus on specific areas [37] to continental-scale studies [25]. NDVI is the most used VI for wheat yield [38], to name a few: [23,24] in China, [36] in Uruguay, [37] in the Canadian prairies, [39] in Grosseto and Foggia, Italy, and [40] in China. The number of studies that use other VIs, such as EVI2 and MTCI, is lower. EVI2 is used by [1] in the region of Punjab and Haryana in India, and MTCI by [11] in South Dakota, USA, [41] in Henan Province, China, [42] in Southern Sweden, and [43] in Iran. Selecting a VI during the process of modeling wheat production and yield may impact the models' performance. NDVI is the normalized difference between red and near-infrared (NIR) bands, and it enables the minimization of noise caused by cloud shadows or topographical effects in the isolated spectral bands [44]. NDVI tends to become more saturated than EVI2 in regions with a high level of biomass while also reducing the influence of the atmosphere and the ground [45]. Other studies, such as [46,47], concluded that EVI2 is more sensitive to variations in vegetative species. Since the launch of Sentinel-2,

it has been possible to calculate various VIs at a greater spatial resolution (10 m). With Harmonized Landsat Sentinel (HLS), both datasets have been normalized, and images are provided approximately every three days [48], which may prove to be useful in areas without much cloud cover, i.e., the Mediterranean.

The methodology of our study is a modified version of that proposed by [25], where they estimated wheat production on a European scale, including 105 NUTS-2 from 19 countries. NUTS are statistical territorial units established by the EU to divide the territory, ranging from level 1 (larger regions) to level 3 (smaller regions). In Spain, there are 7 NUTS-1 regions, 19 NUTS-2 regions, and 59 NUTS-3 regions. A wheat cropland mask or a specific time window adapted to the crop calendar of each NUTS-2 was not considered in the referred work. Wheat croplands were obtained from CORINE Land Cover (CLC) maps [49] and wheat time windows from [50]. In [25], they used a decadal product from 1999 to 2011 of NDVI and FAPAR from the SPOT VEGETATION instrument at 1 km spatial resolution for those NUTS-2 in Europe where official statistics were available. Model calibration was conducted over the BELMANIP2 sites, a network of sites aimed at obtaining a good representativity of the surface types and conditions [51] and preprocessed as described in [52]. In [25], they used the Partial Least Squares Regression method to fine-tune a statistical model for each NUTS-2.

The aim of this study is to develop a suitable yield and production model for Europe using Spain as a test area. The main objective of our study is to build models to estimate both wheat production and yield in Spain based on GS and NDVI, EVI2 and MTCI time series, and to identify the VI with the best performance to implement it at an international scale in places where there are reliable GS about yield and production, as the European Union. Specific objectives include the evaluation of models created for different regions of Spain (NUTS-2) with a special focus on their precision and their correlation with GS, as well as evaluating the impact that wheat irrigation has in yield models. Unlike the study by [25], this work will employ various VIs at a higher spatial resolution (250 and 300 m), and thus a working scale that is 16 or 11 times more detailed; customized time intervals for the agroclimatic characteristics of each region in order to find the annual maximum; and annual masks exclusively for wheat instead of using the CLC arable land category, which also includes other rotation crops, e.g., grain, legumes, vegetables, industrial crops, abandoned agricultural land, and flowers [53]. Lastly, we will evaluate the proposed improvements as regards the application of the methodology in [25] using a CLC wheat mask. Although it is very common in the literature to find studies about wheat production or wheat yield separately, it is not so common to find one that considers both and their different performance. The use of three different VIs adds interest to the above-mentioned, especially by including MTCI, which is much less used than NDVI or EVI2. Therefore, it is the combination of the above-mentioned factors: different indices, Geographic Information System for the CAP (GISCAP) masks (when compared to CLC), and a further evaluation of the reliability of the statistics provided by different regions in Spain. Differences between different NUTS-2 and the effect of irrigated wheat in yield models are also studied. We chose this methodology because, despite its apparent simplicity, it can be used to analyze which of the three VIs yields models with better performance. The limited number of observations prevents the use of other methods that require a larger number of observations. Additionally, comparing different VIs, spatial masks, and official sources requires the ability to visualize each value and identify anomalies.

## 2. Materials and Methods

Wheat cultivation was modeled for five NUTS-2 in Spain: Castile and Leon, Andalusia, Catalonia, Aragon, and Extremadura (Figure 1). The five NUTS-2 are among the six largest in the country, together comprising 303,290 km$^2$ or 60% of Spain's territory. Furthermore, they are among the six with the largest area dedicated to wheat cultivation, which, in 2020, comprised 1,482,155 hectares (ha) of wheat or 77.62% of all wheat surface area in Spain [54].

Castile-La Mancha was the third largest NUTS-2 by area and wheat croplands, but it was not included because the regional administration did not provide CAP data.

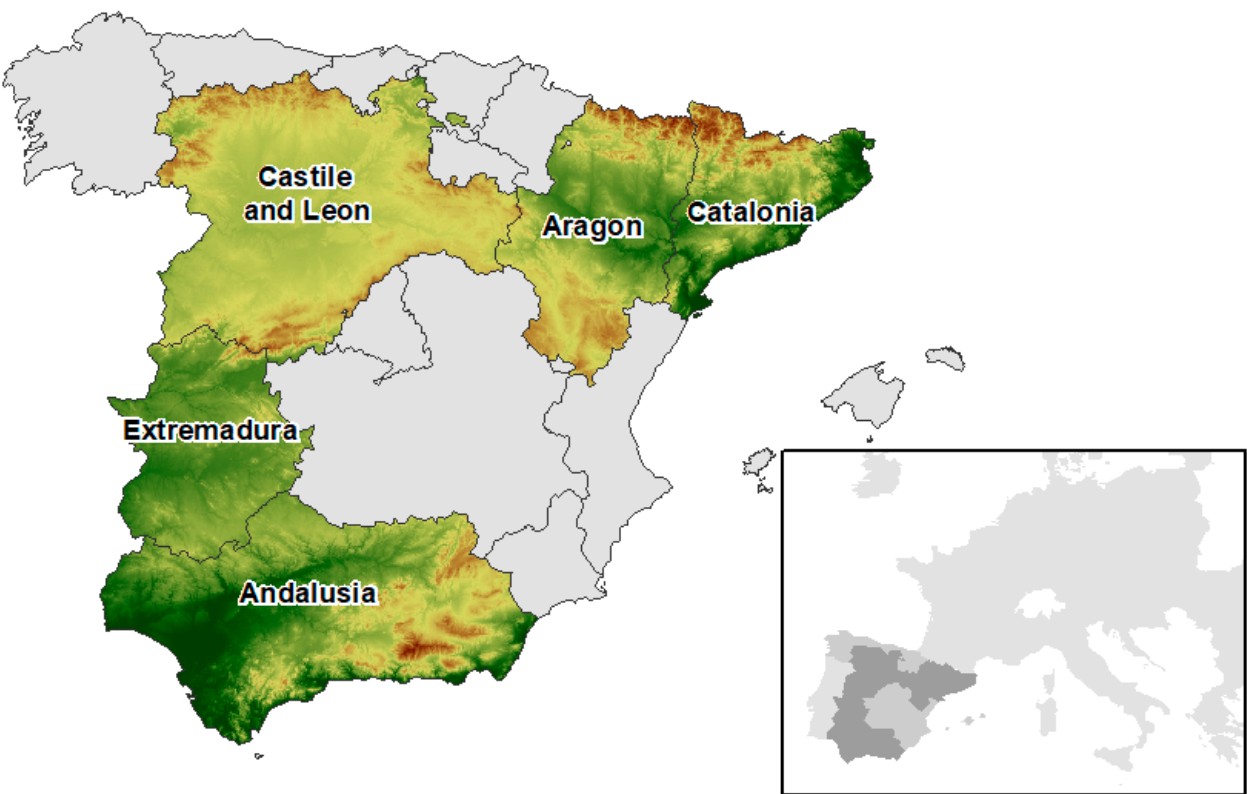

**Figure 1.** Location of the NUTS-2 used in the study.

Several data sources were used to build the models: (i) crop type planted by farmers, as reported in the CAP; (ii) the spatial database from the GISCAP database, featuring the limits for cultivated parcels in Spain, which can be compared to CAP data using numeric codes; (iii) GS for annual yield and production in NUTS-3; (iv) NVDI and EVI2 time series at 250 m spatial resolution for the period between 2006 and 2016, obtained from 8-day composites of surface reflectance images (MOD09Q10)—MODIS is a National Aeronautics and Space Administration (NASA) instrument with high radiometric sensitivity that features 36 bands with a spectral resolution ranging from 0.62 to 14.4 μm [55]; (v) MTCI time series at a spatial resolution of 300 m for the period between 2006 and 2011, obtained from weekly composites from MERIS level 2—MERIS is a European Space Agency (ESA) instrument with 15 spectral bands whose centers range from 412.5 to 900 nm [56]; and (vi) CLC for 2006 and 2012. The study relied on various official data sources, and the accuracy and reliability of these sources significantly influenced the performance of this study. In the case of CAP data, farmers reported the crops cultivated in their plots to the regional administration for each agricultural campaign. Furthermore, field inspections were conducted on approximately 5% of the plots to verify the accuracy of these declarations. The financial aid that farmers receive depends on these declarations. Official yield and production statistics in Spain were compiled by regional administration officials. While there was no documentation available regarding the quality assessment of this data or the accuracy measures of this statistical operation, it had to adhere to the guidelines outlined in the European regulation. These guidelines encompassed standards related to relevance, precision, timeliness, accessibility, clarity, comparability, and consistency [54]. CLC is a pan-European land cover and land use inventory with 44 thematic classes produced via photointerpretation, with a minimum mapping unit of 25 hectares (ha) for areal phenomena and a minimum width of 100 m for linear phenomena [53].

Our methodology consisted of the following steps: (i) locating wheat parcels each year; (ii) creating NDVI, EVI2, and MTCI time series and accumulating their maximum values, plus four subsequent composites for each year; (iii) spatial aggregation of collected VIs values on a provincial (NUTS-3) level and by year; and (iv) linear regression of aggregated accumulated data for EVI2, MTCI, NDVI, and GS for both production and yield (Figure 2). Furthermore, we replicated the methodology proposed by [25] using CLC masks for comparative purposes.

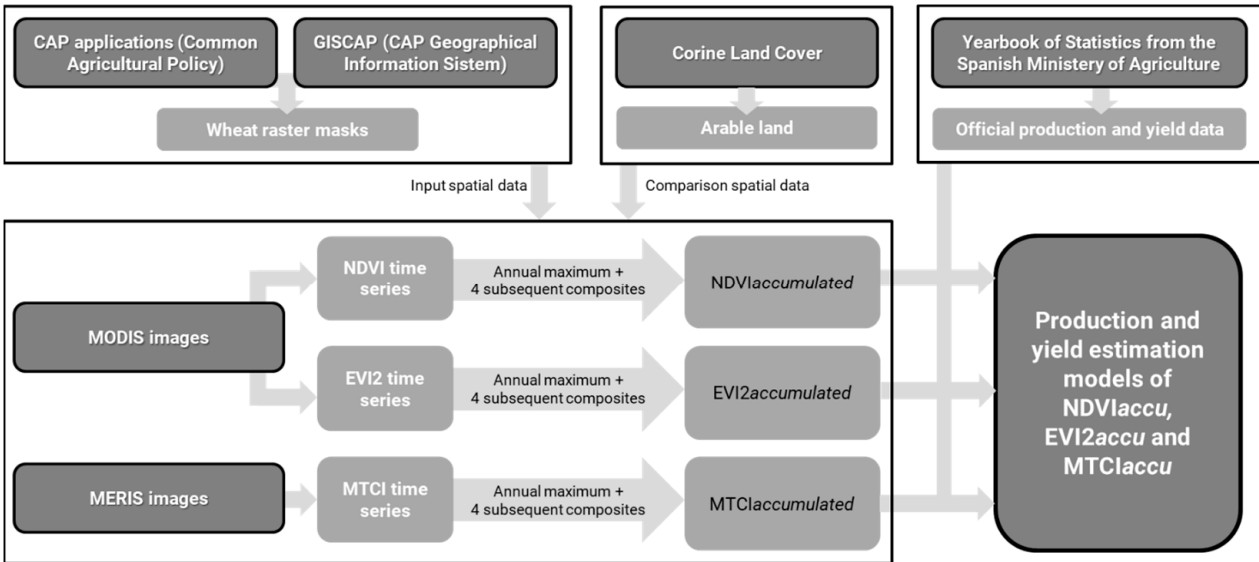

**Figure 2.** Methodology summary.

The location of wheat parcels was obtained using CAP and GISCAP data for each year. Two annual masks were created for each NUTS-2: one based on MODIS and the other based on MERIS. NDVI, proposed by [57], was formulated as follows:

$$\text{NDVI} = \frac{(\text{NIR} - \text{RED})}{(\text{NIR} + \text{RED})}$$

where NIR and RED were the reflectance in the near-infrared (NIR) and red bands. EVI2, as proposed by [58], was formulated as

$$\text{EVI2} = 2.5 \frac{(\text{NIR} - \text{RED})}{(\text{NIR} + 2.4 \times \text{RED} - 1)}$$

where NIR and RED were the reflectance in the near-infrared (NIR) and red bands. In order to calculate NDVI and EVI2, the surface reflectance images (MOD09Q1) [55] were mosaicked and reprojected in sinusoidal projection mapped to geographical coordinates and datum WGS84. Product quality flags were also used to discard pixels where atmospheric correction could not be performed. Afterward, time series were generated for each index, taking into account 506 weekly composites (8 days) across the 11-year study period. MTCI, proposed by [11], was formulated as:

$$\text{MTCI} = \frac{(\text{Rband10} - \text{Rband9})}{(\text{Rband9} + \text{Rband8})}$$

where Rband10 represented the center of MERIS band 10 (753.75 nm), Rband9 represented the center of MERIS band 9 (708.75 nm), and Rband8 represented the center of MERIS band 8 (681.25 nm). The MTCI time series was obtained using the MERIS MTCI Level 2 product [56] by calculating the arithmetic mean for the 8-day periods. Discrete Fourier transform (DFT) was used as a smoothing function to split up the curve into a series of

sinusoidal waves of varying frequencies. We used the first four harmonics in order to reconstruct the smoothed curve using the inverse Fourier transform technique [59], which has been tested as an optimal smoothing function [60]. Finally, we filled the remaining gaps in the time series by linear interpolation based on the immediately preceding and following composite values.

The study used VIs at different spatial resolutions (NDVI and EVI2 from MODIS at 250 m; MTCI from MERIS at 300 m) at 6.25 ha per MODIS pixel and 9 ha per MERIS pixel. However, these variations in area did not have a big impact on the results, as the study was conducted at a national level. In all NUTS-2, the number of MODIS pixels was larger than MERIS pixels, although MERIS pixel area was higher (Table 1). We selected rainfed wheat parcels that were larger than 4 hectares (ha) (Figure 3). Choosing a minimum parcel size of 6.25 or 9 ha would have drastically reduced the number of available parcels. In order to eliminate those pixels where wheat was not predominant, two further conditions were imposed: (i) wheat parcels with at least 90% of their area effectively occupied by wheat and (ii) at least 50% of the pixel area was occupied by one or more parcels that met the previously mentioned criteria. With regard to requirement (i), farmers had to state which percentage of the wheat parcel was effectively cultivated with wheat (not including roads, constructions, or other crops in the same parcel).

**Table 1.** Mean and standard deviation of the size of selected wheat parcels in ha, number of MERIS and MODIS pixels, and MODIS and MERIS area in km$^2$. Data show mean values for the time series.

| | Mean Wheat Parcel Size (ha) | Standard Deviation of Wheat Parcel Size (ha) | Number of MODIS Pixels | MODIS Pixel Area (km$^2$) | Number of MERIS Pixel | MERIS Pixel Area (km$^2$) |
|---|---|---|---|---|---|---|
| Andalusia | 13.4 | 15.9 | 22,737.2 | 1421.1 | 20,543.3 | 1848.9 |
| Aragon | 8.9 | 8.9 | 59,288.3 | 3705.5 | 50,929.2 | 4583.6 |
| Castile and Leon | 7.3 | 5.6 | 98,681 | 6167.6 | 91,323.5 | 8219.1 |
| Catalonia | 7.1 | 4.2 | 13,670.7 | 854.4 | 13,224.5 | 1190.2 |
| Extremadura | 11.8 | 13.2 | 8660.8 | 541.3 | 8433.8 | 759 |

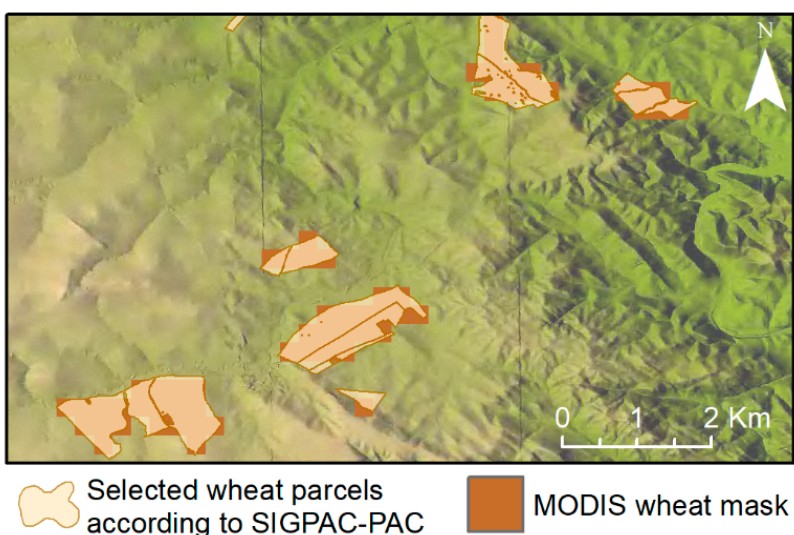

**Figure 3.** Example of parcel and pixel overlapping in Extremadura for the year 2006.

Time series information was extracted for the selected pixels, and the mean was calculated in order to create a temporal profile for each NUTS-2. Specific time windows were established according to each NUTS-2 agroclimatic characteristic to find the maximum VI value (Table 2). It was made by selecting the range of composites where wheat pixels reached the first maximum value. It reduced the impact of those parcels where summer

crops were also cultivated. The climatic differences, agricultural practices, or soil types between the different regions of Spain were very high, which was why, in some regions, the maximum values of the VI were reached more than a month apart. Furthermore, the differences were also notable even within the same region, and the time windows were larger, too. According to some studies [41,61], a wheat period that best explains future wheat production corresponds to the moment when maximum VI values are reached. Following the proposals made in other studies [1,62,63], we used the maximum value and the 30-day period immediately after the summation of the composite where the maximum VI value was reached for each agricultural campaign and year plus the four subsequent composites.

**Table 2.** Time window to find the maximum VI value.

|  | NDVI (DOY) | EVI2 (DOY) | MTCI (DOY) |
| --- | --- | --- | --- |
| Andalusia | 40–136 | 40–136 | 40–136 |
| Aragon | 40–216 | 24–208 | 16–192 |
| Castile and Leon | 32–232 | 72–208 | 8–200 |
| Catalonia | 8–176 | 24–192 | 48–192 |
| Extremadura | 8–176 | 8–152 | 16–144 |

Production and yield data for the NUTS-3 level were merged into NUTS-2. Two values were obtained for each of the 26 NUTS-3 (9 in Castile and Leon, 8 in Andalusia, 4 in Catalonia, 3 in Aragon, and 2 in Extremadura), VI and year: the summation and the mean of each VI for the period that includes the maximum IV value and the following month. Datasets for NDVI and EVI2 in Spain models used 286 points (99 in Castile and Leon, 88 in Andalusia, 44 in Catalonia, 33 in Aragon, and 22 in Extremadura). Datasets for MTCI in Spain models in Spain used 156 points (54 in Castile and Leon, 48 in Andalusia, 24 in Catalonia, 18 in Aragon, and 12 in Extremadura), as MERIS time series were shorter than MODIS time series. Production is a cumulative and absolute variable, as referred to in GS, so it was linked to the summation of the referred period, which is also an absolute variable. Yield is a relative variable defined as production per unit of space and therefore linked to the mean of the referred period, which is also a relative variable, as in other studies [1,62,63]. Thus, the impact would not be very high in the case of data inaccuracies, e.g., with wheat mask or with GS. Linear regression analyses were then carried out for production and yield. These analyses yielded a coefficient of determination ($R^2$) and a normalized relative error (NRE) for the average of the dependent variable, which was also performed in other studies [64,65]. Irrigation could have an important role in wheat yield. Our approach was repeated to include not only rainfed wheat but also irrigated wheat, analyzing how models are affected by this aspect.

In [25], they noted three possible improvements for their study: (i) the use of crop masks due to the inclusion of other crops in the arable land category and using a greater spatial resolution; (ii) the use of longer time series, as the 13-year time series that they used was due to limited data availability in the SPOT Vegetation series; and lastly (iii) the use of specific wheat calendars for each region. Improvements (i) and (iii) were implemented in this study, repeating the previously mentioned methodology but including masks from CLC arable land [49] instead of GISCAP-CAP masks. However, CLC masks were made every 6 years, and they were only available in 2006 and 2012 during the study period. Therefore, the model was calculated for Spain, and not by NUTS-2, due to the lack of values, as each data point in the regression analysis represents a NUTS-3 (26 in total) per year. Lastly, linear regression analyses were also performed between the area reported by GS and the area calculated based on GISCAP-CAP and CLC to evaluate their differences.

## 3. Results

### 3.1. Analysis of VIs' Time Series

The minimum value in the time series occurred at the end of summer, after wheat harvest, and prior to the next sowing [66]. As seeds began to germinate, which coincided with a decrease in temperatures and the arrival of autumn precipitation, the values of the VIs began to increase until the period between February and May—depending on the climate conditions in each region—with the maximum coinciding with the sprouting phase. From this moment onward, wheat began to mature while subsequently drying out due to the increasing temperatures in spring, thus causing a dramatic decrease in VIs values. Harvesting began at the start of summer, around the end of June or the start of July, and a new agricultural cycle for wheat began at the end of summer (Figure 4). NDVI reached its maximum value, on average, on day of the year (DOY) 99, EVI2 on DOY 100, and MTCI on DOY 101. This slight difference led to relatively homogeneous behavior in the post-maximum period overall (NDVIaccum, EVI2accum, and MTCIaccum). The NDVI maximum value usually exceeded 0.6, except in Aragon.

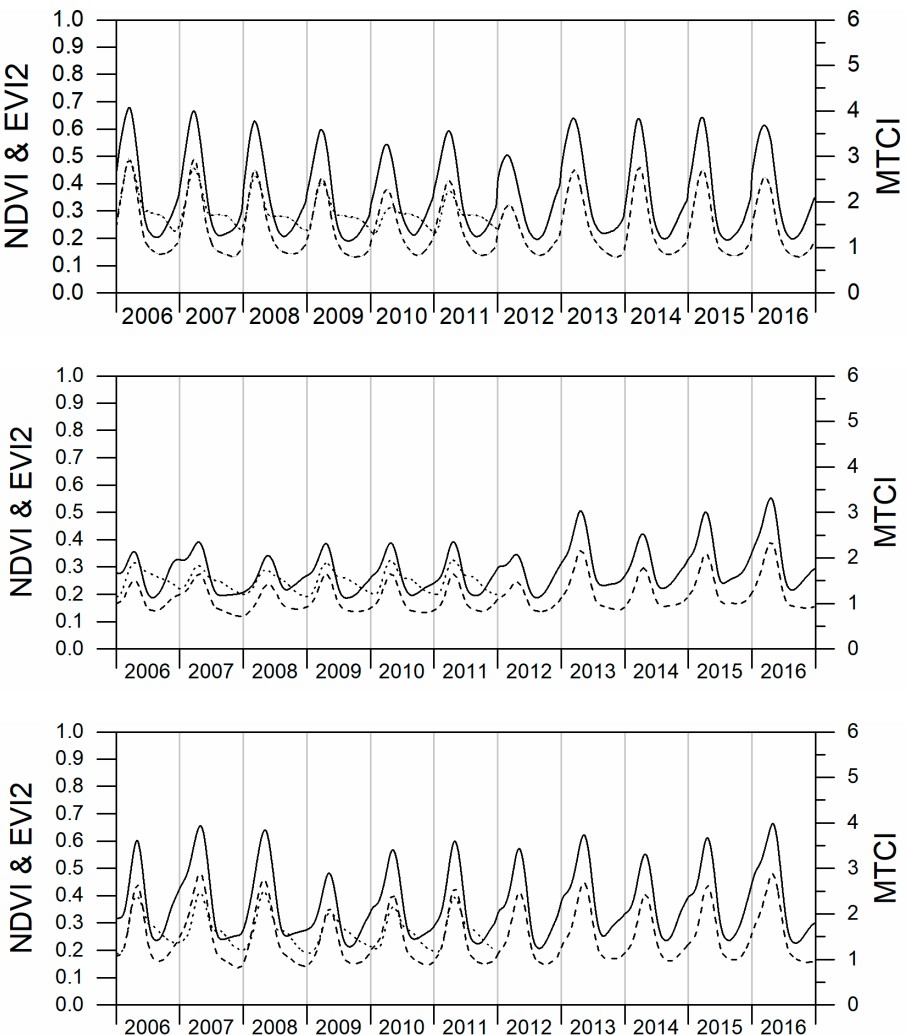

**Figure 4.** *Cont.*

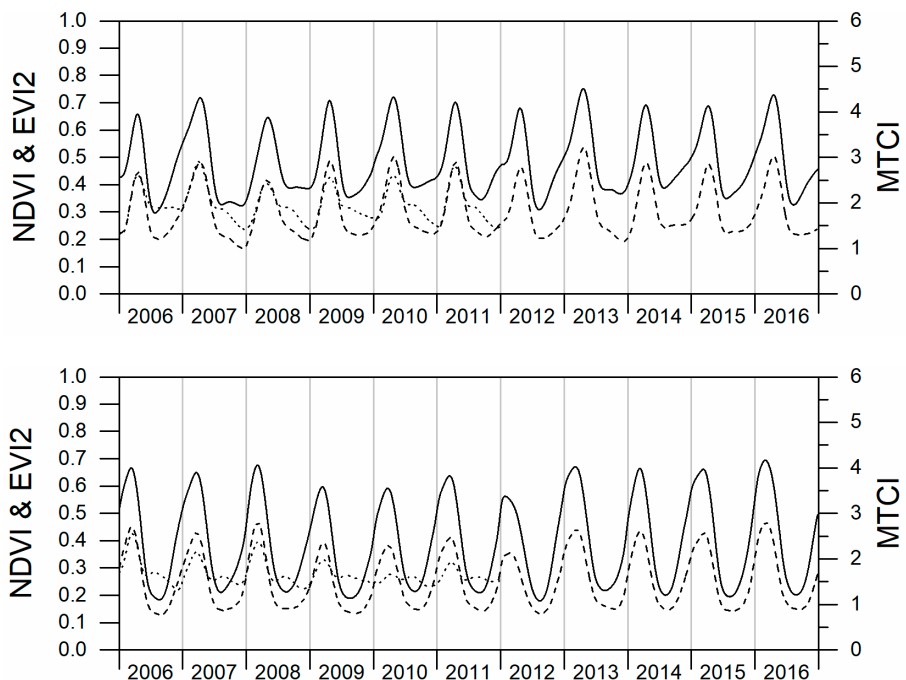

**Figure 4.** Time series for the median values of NDVI (continuous), EVI2 (dashed), and MTCI (dotted) in the five NUTS-2 during the 2006–2016 time series. Top to bottom: Andalusia, Aragon, Castile and Leon, Catalonia, and Extremadura.

### 3.2. Analysis of Wheat Masks

There were significant differences in area among the masks used (Figure 5). CLC masks overestimated the wheat area of all NUTS-2 by a factor of 4 and 8 compared to the area provided in GS (Table 3). GISCAP-CAP masks gave the closer result to GS data: 88.57% in 2006 and 117.18% in 2012 for all five NUTS-2 (Table 3).

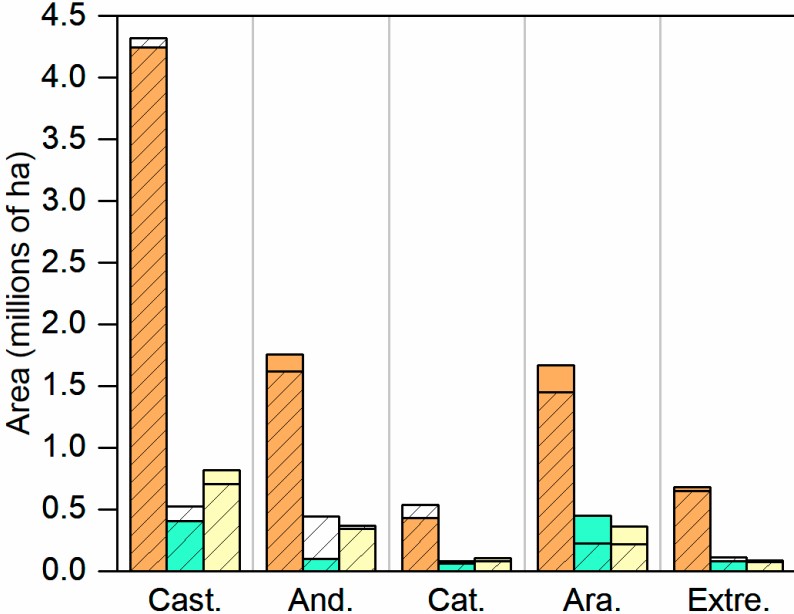

**Figure 5.** Area comprising wheat, by NUTS-2. Cast., Castile and Leon; And., Andalusia; Cat., Catalonia; Ara., Aragon; Extre., Extremadura. Orange, CLC; Green, GISCAP-CAP; Yellow, GS. Color, 2006 values; Shaded, 2012 values.

**Table 3.** Surface (%) according to wheat masks from GISCAP-CAP and CLC compared to GS in 2006 and 2012.

| | | Castile and Leon | Andalusia | Catalonia | Aragon | Extremadura | Total |
|---|---|---|---|---|---|---|---|
| 2006 | GISCAP-CAP | 76.83% | 21.83% | 138.97% | 199.79% | 69.94% | 88.57% |
| | CLC | 811.19% | 398.86% | 738.93% | 746% | 618.00% | 704.98% |
| 2012 | GISCAP-CAP | 116.28% | 93.84% | 135.19% | 163.43% | 87.97% | 117.18% |
| | CLC | 613.60% | 442.54% | 692.07% | 656.52% | 778.56% | 590.71% |

Arable land from CLC comprised 87.79% of the total area included in the study, wheat from GISCAP-CAP comprised 2.77%, and the remaining 9.44% is where overlap occurred between the two datasets (Figure 6). This surface disparity highlights the importance of mask selection. Aragon had the greatest number of matching pixels according to both sources (19.42%), although the group of pixels that only included CLC was still the largest.

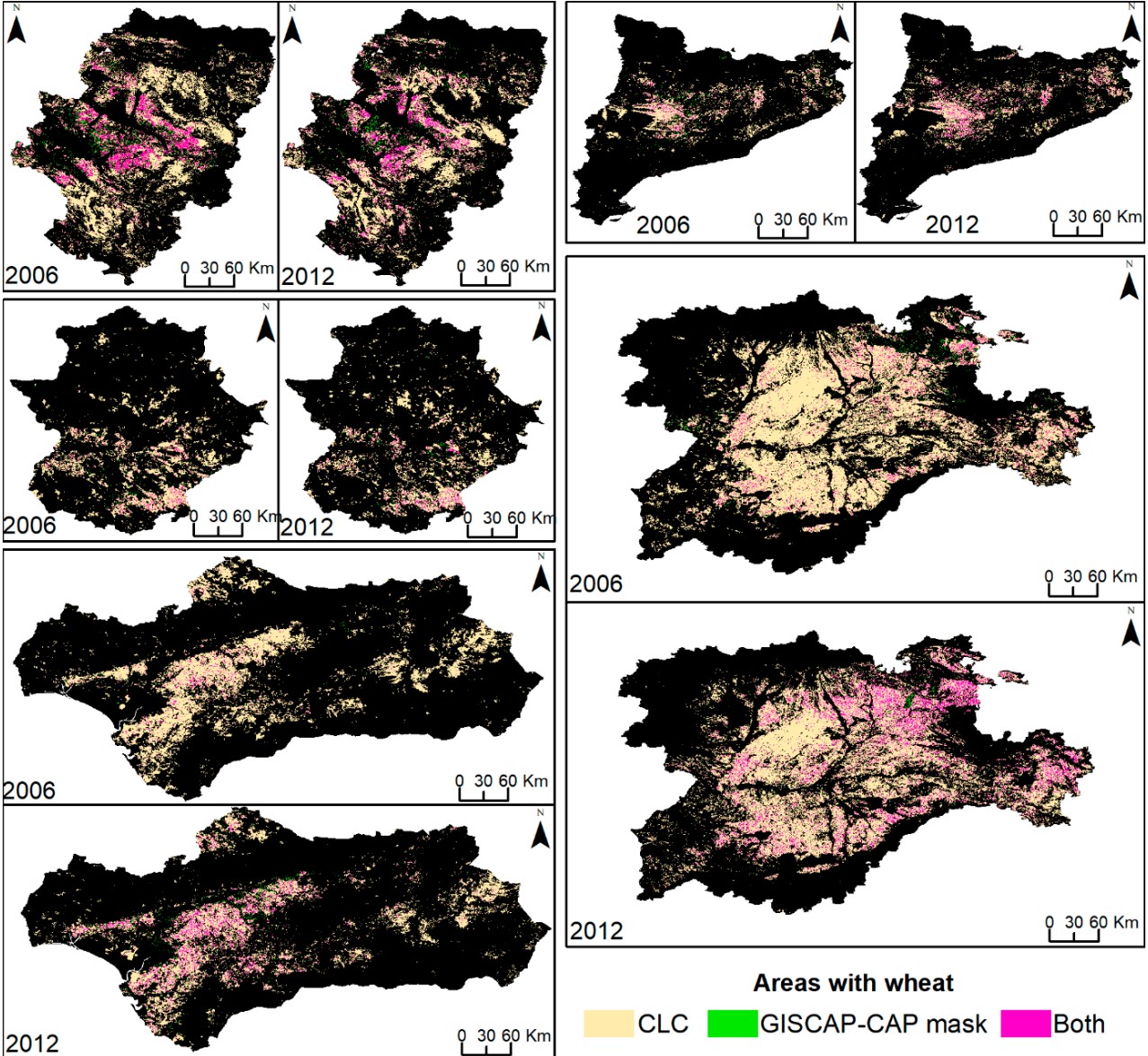

**Figure 6.** Areas with wheat according to CLC and PAC-SIGPAC wheat masks in 2006 and 2012. Top to bottom, left to right: Aragon, Extremadura, Andalusia, Catalonia, Castile, and Leon.

Linear regression analysis between the area reported by GS and the area calculated based on MODIS and CLC masks revealed a positive correlation, with the exception of some years in Andalusia and Aragon (Figure 7). A similar pattern was observed when correlating the values calculated for production and area. The fact that CLC data were only available for 2006 and 2012 denoted a lack of precision when correlating this cover with GS area and GS production. Aragón and Andalusia were the NUTS-2 that had production models with the lowest $R^2$ values and the highest NRE values.

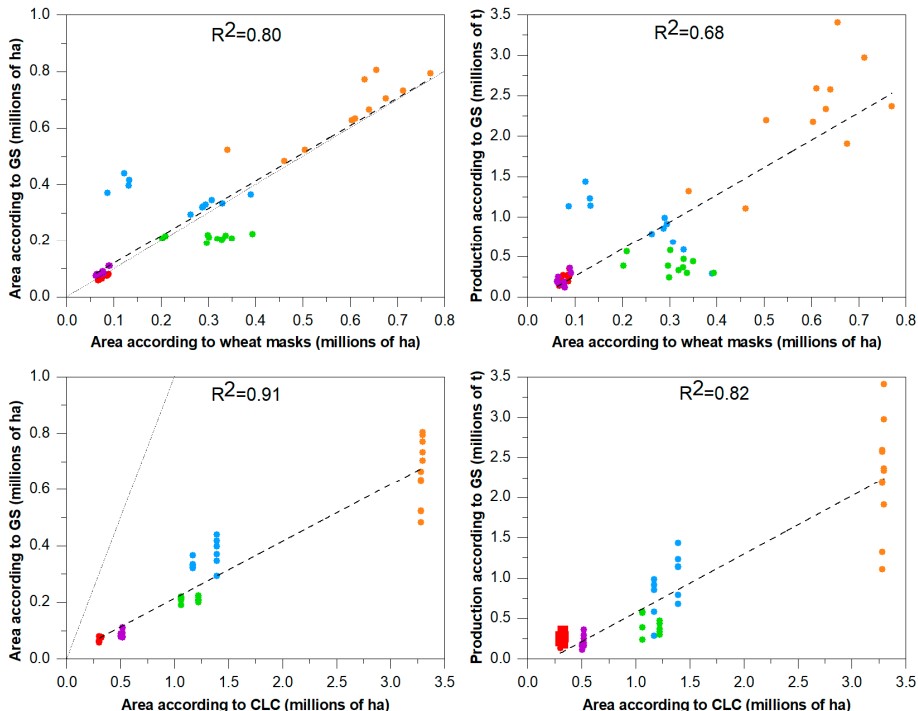

**Figure 7.** From left to right, top to bottom. Area according to wheat masks vs. GS. Area according to wheat masks vs. production from GS. Area according to CLC vs. GS. Area according to CLC vs. production from GS. Each data point represents a NUTS-3 and year. Orange represents values for Castile and Leon; Blue, Andalusia; Red, Catalonia; Green, Aragon; Purple, Extremadura. The dashed line represents the linear equation; the dotted line represents a 1:1 ratio. Bottom figures use area according to CLC with data only for 2006 and 2012.

### 3.3. Modeling Production and Yield

All production models were significant at the 95% confidence level (Table 4), but results differed between each NUTS-2. On one hand, models for Castile and Leon had a greater forecasting capability, with $R^2$ values that exceeded 0.96 and NRE values below 18% for all VIs. On the other hand, the models in Andalusia had $R^2$ values between 0.47 for EVI2 and 0.68 for MTCI and NRE values close to or greater than 50%. Production models in Extremadura and Catalonia also had a high forecasting capability, with $R^2$ values for all VIs higher than 0.8 or even 0.9, except for MTCI in Catalonia. Aragon showed lower $R^2$ values, only surpassing 0.8 for MTCI. MTCI was the VI that created production models with the best forecasting capability in Castile and Leon, Andalusia and Aragon, and EVI2 in Extremadura and Catalonia. Models for Spain were lower than for the majority of NUTS-2: EVI2 had an $R^2$ of 0.75 and an NRE of almost 33%, NDVI 0.71 and 36.71%, respectively, and MTCI 0.62 and 41%, respectively. Models that used yield instead of production obtained lower $R^2$ values and higher NRE values (Tables 4 and 5) (Figure 8). Regarding yield models, MTCI produced the models with the best forecasting capability in Andalusia and Extremadura, EVI2 in Aragon and Catalonia, and NDVI in Castile and Leon. Aragon showed the highest forecasting capability with an $R^2$ and the lowest NRE with values of 0.85 and 10.57%, respectively, with EVI2, and Extremadura the lowest with an

$R^2$ of 0.01 and an NRE of 24.54% with NDVI. None of the three VIs were significant at the 95% confidence level for yield models in Extremadura, nor in Catalonia for MTCI. Models for Spain showed a low forecasting capability with an $R^2$ of 0.44 and an NRE of 21.88% for EVI2, 0.38 and 22.76%, respectively, for NDVI and 0.37 and 21.23%, respectively, for MTCI.

**Table 4.** $R^2$ and NRE values for production models. Values with an asterisk (*) are significant at the 95% confidence level. Production models use VIaccu sum.

|  | $R^2$ | | | NRE | | |
|---|---|---|---|---|---|---|
|  | NDVI | EVI2 | MTCI | NDVI | EVI2 | MTCI |
| Andalusia | 0.47 * | 0.47 * | 0.68 * | 54.63% | 52.79% | 49.35% |
| Aragon | 0.77 * | 0.78 * | 0.86 * | 25.29% | 23.53% | 17.63% |
| Castile and Leon | 0.96 * | 0.96 * | 0.98 * | 17.71% | 17.88% | 14.72% |
| Catalonia | 0.82 * | 0.84 * | 0.69 * | 14.43% | 13.33% | 20.40% |
| Extremadura | 0.85 * | 0.94 * | 0.92 * | 24.92% | 24.68% | 22.37% |
| Total | 0.71 * | 0.75 * | 0.6 * | 36.71% | 32.99% | 41.07% |

**Table 5.** $R^2$ and NRE values for yield models. Values in round brackets include irrigated wheat. Values with an asterisk (*) are significant at the 95% confidence level. Models for yield use VIaccu mean.

|  | $R^2$ | | | NRE | | |
|---|---|---|---|---|---|---|
|  | NDVI | EVI2 | MTCI | NDVI | EVI2 | MTCI |
| Andalusia | 0.56 * | 0.54 * | 0.61 * | 18.86% | 19.55% | 13.74% |
|  | (0.61) | (0.59) | (0.66) | (19.16%) | (19.83%) | (14.17%) |
| Aragon | 0.81 * | 0.81 * | 0.57 * | 13.04% | 10.57% | 13.12% |
|  | (0.78) | (0.85) | (0.59) | (11.21%) | (10.57%) | (13.11%) |
| Castile and Leon | 0.59 * | 0.51 * | 0.44 * | 17.47% | 18.94% | 19.95% |
|  | (0.41) | (0.3) | (0.25) | (17.43%) | (18.97%) | (19.83%) |
| Catalonia | 0.25 * | 0.31 * | 0.06 | 13.04% | 12.75% | 18.76% |
|  | (0.26) | (0.39) | (0.1) | (13.19%) | (13.11%) | (18.73%) |
| Extremadura | 0.01 | 0.02 | 0.26 | 24.54% | 24.44% | 16.43% |
|  | (0.01) | (0.01) | (0.25) | (24.48%) | (24.31%) | (16.37%) |
| Total | 0.38 * | 0.44 * | 0.37 * | 22.76% | 21.88% | 21.23% |
|  | (0.36) | (0.38) | (0.32) | (22.79%) | (22.20%) | (20.94%) |

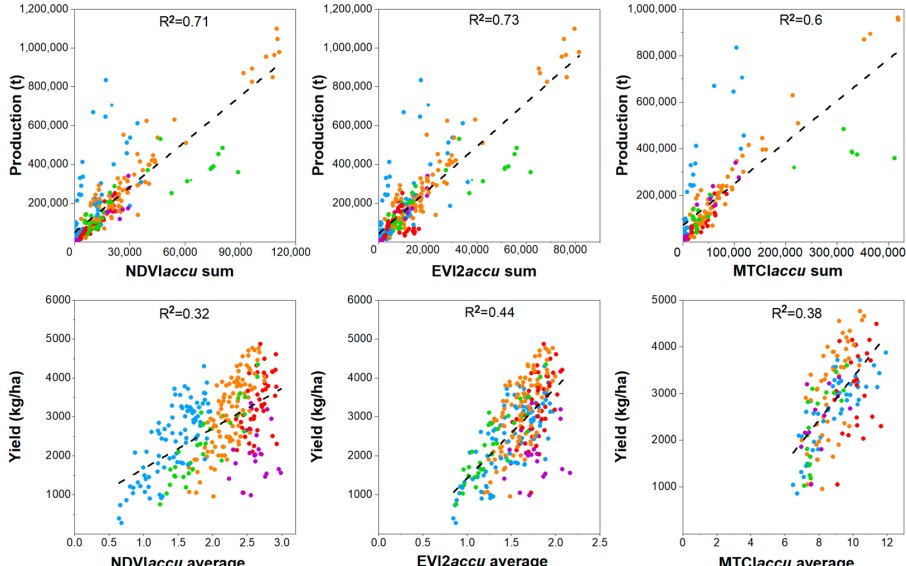

**Figure 8.** Results of regression analysis for Spain. Top plots show production models, and bottom plots show yield models. Each data point represents a NUTS-3 and year. Orange represents values for Castile and Leon; Blue, Andalusia; Red, Catalonia; Green, Aragon; Purple, Extremadura.

Models that used CLC as input data were significant at a 95% confidence level, with the exception of yield models for EVI2 in 2006, despite most of them not achieving an $R^2$ value of 0.6 and none exceeding 0.7 (Table 6). Models that included production as a dependent variable had greater $R^2$ values than those for yield. However, production models generated higher NRE values of around 60%.

**Table 6.** $R^2$ and NRE values for production and yield values in the five regions studied using CLC data. Values with an asterisk (*) are significant at the 95% confidence level.

| | | $R^2$ | | NRE | |
|---|---|---|---|---|---|
| | | Production | Yield | Production | Yield |
| NDVI | 2006 | 0.31 * | 0.05 | 68.01% | 25.98% |
| | 2012 | 0.67 * | 0.59 * | 55.51% | 22.81% |
| | Both | 0.47 * | 0.33 * | 60.71% | 27.20% |
| EVI2 | 2006 | 0.44 * | 0.45 * | 62.39% | 20.65% |
| | 2012 | 0.69 * | 0.53 * | 55.01% | 23.65% |
| | Both | 0.55 * | 0.51 * | 58.01% | 22.34% |
| MTCI | 2006 | 0.55 * | 0.55 * | 59.38% | 17.70% |

## 4. Discussion

The different behavior of the VIs time series might be caused by the following reasons: (i) NDVI saturates fast with crop canopy development, leading to very little change in NDVI when LAI reaches values greater than 3 [19,67]; (ii) NDVI has a confounded response to chlorophyll content and plant area, making it difficult to distinguish changes in plant structure from changes in leaf biochemistry [35,68]; and (iii) NDVI is sensitive to the albedo of bare soil, particularly at low LAI/biomass values [69]. EVI2 showed a similar pattern to NDVI, with both VIs showing a very similar response to seasonal events linked to the phenological characteristics of wheat crops. However, EVI2 showed maximum and minimum values that were lower overall compared to other studies [46,70]. EVI2 values were lower in Aragon compared to the rest of NUTS-2, as well as NDVI. MTCI, which is more sensitive to chlorophyll content [71], also showed a relative maximum during the summer period. This could be due to both the remaining vegetation present after harvesting and other self-sown vegetation, which reached its minimum after farmers had removed vegetation at the start of the new agricultural cycle.

Despite the proximity between NUTS-2, there were considerable differences in the behavior of the VIs. It may be due to differences in altitude, climate, ocean proximity, soil type, or agricultural practices (Figure 9). The maximum occurred in mid-March as a result of milder winters and an earlier increase in temperatures in the two southernmost regions, Andalusia and Extremadura (Figure 10). Furthermore, wheat distribution in these NUTS-2 was primarily concentrated in low-altitude areas, i.e., lower than 250 m in Andalusia and 425 m in Extremadura (Table 7). In the northernmost regions, maximum values were achieved around the end of April or the beginning of May. There, the average wheat altitude was higher, especially in Castile and Leon. It was common for VI values to increase at the start of spring in these northern NUTS-2, with an intermediate period of stabilization preceding another increase where the maximum was reached. Inter-annual differences in climate had an impact on rainfed crops, as temperatures and water availability were both determinant factors of production and yield for the season. Optimal climate conditions for wheat growth occurred in years such as 2013 and 2016. This was reflected in higher VI values that lasted longer over time. However, the opposite behavior was observed in 2012 and 2014 (Figure 4).

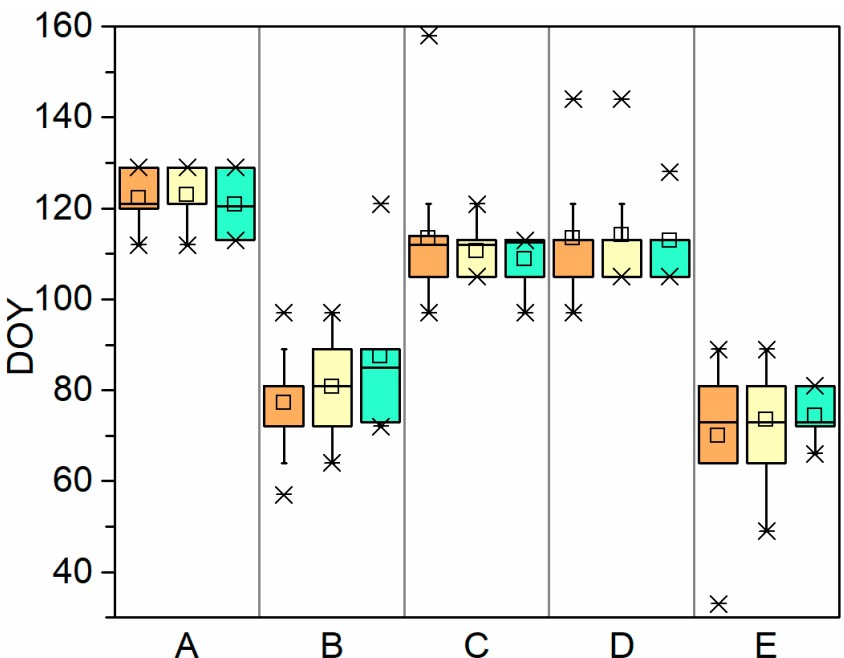

**Figure 9.** Average DOY when the maximum value was achieved during the study period, grouped by NUTS-2 and VI. A, Castile and Leon; B, Andalusia; C, Catalonia; D, Aragon; E, Extremadura. Orange, NDVI; Yellow, EVI2; Green, MTCI.

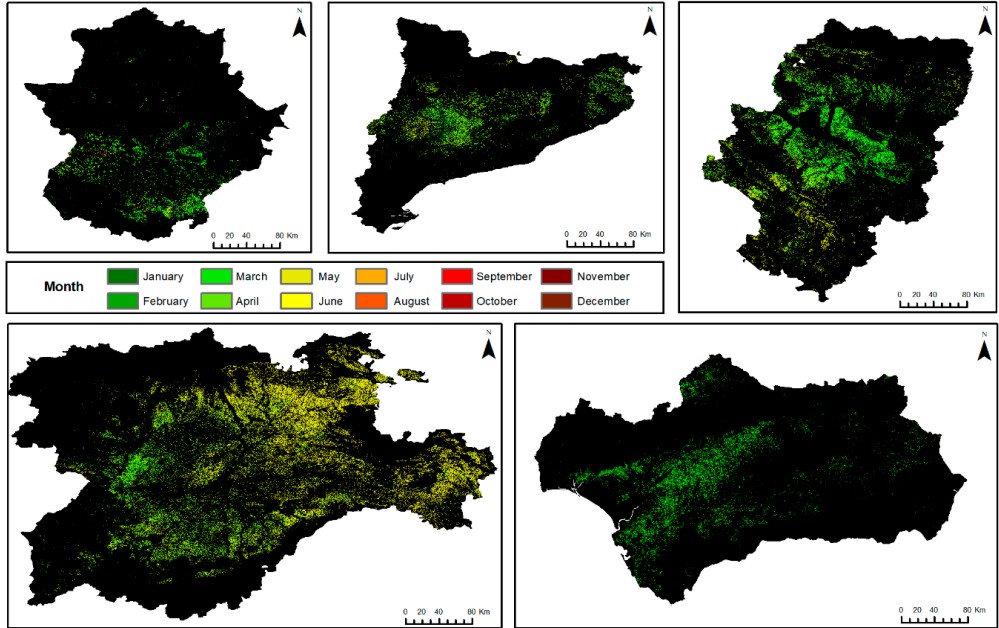

**Figure 10.** Month where each wheat pixel reached the NDVI maximum value. For pixels with wheat for several years, mean value is represented.

**Table 7.** Median, lowest, and highest altitude (m) of wheat parcels by NUTS-2.

|  | Andalusia | Aragon | Castile and Leon | Catalonia | Extremadura |
|---|---|---|---|---|---|
| Mean altitude | 248.26 | 526.86 | 872.23 | 530.91 | 424.25 |
| Lowest altitude | 1 | 120 | 377 | 1 | 149 |
| Highest altitude | 1911 | 1815 | 1485 | 1649 | 684 |

According to GISCAP-CAP masks, wheat was only grown in most pixels for a year due to the agricultural practices inherent to wheat and inter-annual crop rotation (Figure 11). Moreover, its profitability was lower compared to other crops, which explains why it has only grown on the same parcel for a few years. Aragon was the only location where a significant number of pixels were found to contain wheat over several years, in the central area in particular.

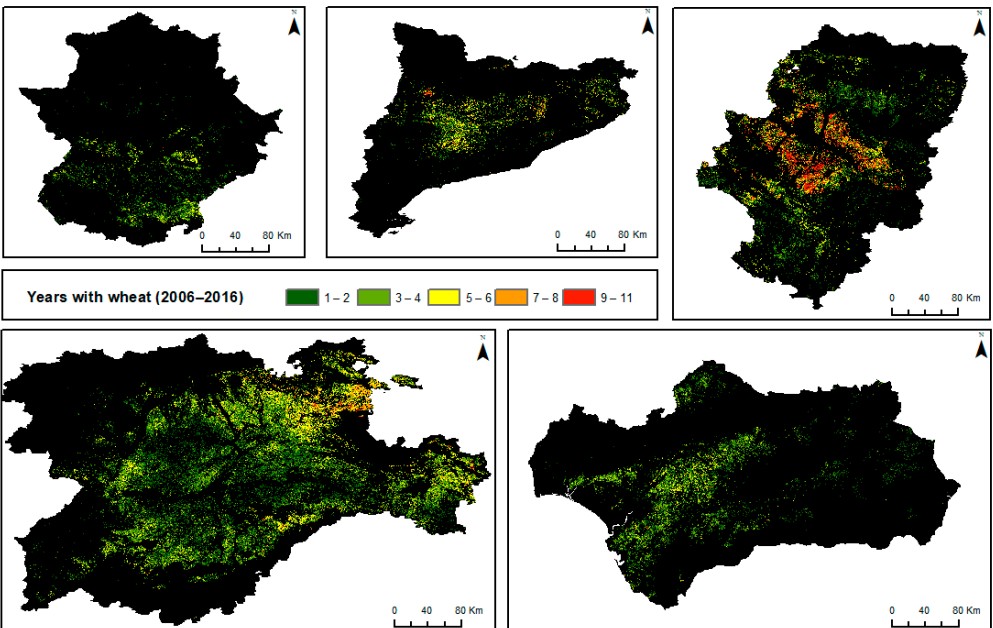

**Figure 11.** Number of years that each pixel represented wheat in the time series between 2006 and 2016. Left to right, top to bottom: Extremadura, Catalonia, Aragon, Castile and Leon, Andalusia.

Remote sensing could enable to obtain annual masks without relying on land covers such as CLC or CAP data in regions of the world where they are not available. Given that crops have varying spectral responses, these differences enable us to extract phenometrics about vegetative development, such as leaf onset, end of senescence, dates of maximum or minimum values, etc., which can be used for crop classification [44]. Some previous studies used MODIS VI products (NDVI and EVI) for crop classification [46,72]. The fact that these 8-day composites are created based on daily information helps to avoid any data gap. However, due to climate conditions in Spain, which has less cloud cover than other areas of Europe, we could be able to evaluate the use of data from Sentinel-2 (Sentinel-2A from 2015 and Sentinel-2B from 2017). It could be possible to calculate the missing values using smoothing, such as DFT, Savitzky–Golay filter, or the double-logistic function, enabling spatial resolutions of 10 m. Nonetheless, the short length of the available time series made it difficult to carry out these types of studies, as they largely benefit from longer time series.

Regarding the performance of production models, results for NDVI and EVI2 were, in general, quite similar, which is in line with other studies that also used NDVI and EVI2 from MODIS, such as [46], which showed a correlation of 96% between both VIs, although NDVI values were higher. According to [31], NDVI also outperformed EVI2. Previous studies that also used MODIS VI data [1,7,31,63] obtained similar results, with $R^2$ values of around 0.7 [73] and 0.6 for yield and 0.86 and 0.92 for production [1]. Other studies compared wheat yield with VIs obtained using the Advanced Very High Resolution Radiometer (AVHRR) instrument, obtaining $R^2$ values of 0.88 [74]; between 0.47 and 0.8, depending on the climate zone [65]; and 0.28 [7]. The results of [41] showed that yield models with MTCI had an $R^2$ value of 0.85, and with NDVI, $R^2$ was 0.56. The lower forecasting capability of yield models could be partly due to the fact that area is implicitly included in the sum of pixel values. Thus, the larger the cultivated area, the higher the number of pixels and the amount of production. This was not the case with yield because it used the mean

value [1,63]. Moreover, the lowest $R^2$ values in yield models could be in part because it was not possible to differentiate between different wheat varieties, e.g., durum wheat and common wheat [26]. Results suggested that they were not highly affected by the presence or absence of irrigation, with a similar performance in general (Table 5).

The differences in model performance across the various NUTS-2 regions may be partly attributed to the very different characteristics that existed in Spain. It included altitude, proximity to the coast, temperature, precipitation, soil characteristics, agricultural practices, and wheat varieties present in Spain. Significant intra-regional disparities may also contribute to the decreased accuracy of models in certain NUTS-2 regions. Therefore, modeling the production and yield of wheat or any crop represented an extra challenge in those areas with very different characteristics, as the results may not be as accurate. Moreover, NUTS-2 with a smaller wheat area could be more sensitive to outliers, e.g., Castile and Leon, the NUTS-2 with the largest wheat area and production had production models with the highest $R^2$ values and the lowest NRE values, while Extremadura, the NUTS-2 with the smallest wheat area and production, had yield models with a low forecasting capability. The different results for each NUTS-2 could also be explained by the number of NUTS-3 in which each NUTS-2 was divided. Castile and Leon had nine NUTS-3, Andalusia had eight, Catalonia had four, Aragon had three, and Extremadura had two. Therefore, any outlier could have a big impact on NUTS-2 with less NUTS-3. The difference in climate could also affect, as stated by [46].

Adverse weather events that could take place after the post-maximum period used in this study (cold or heat waves, storms, hail, droughts, or floods) could potentially have an impact on models, given that production and yield would tend to be overestimated. The model could also underestimate production when climate conditions were more favorable for wheat growth. In [25], they noted how the models showed a higher number of errors in years where temperatures and precipitations deviated more from average climate conditions, e.g., 2003. Other authors, such as [75], showed that there was a lower correlation between crop yield and NDVI time series in years with greater humidity. They also showed that models comparing wheat production to VIs had higher $R^2$ values in semi-arid areas where precipitation levels did not exceed 600 mm. In [27], they noted that higher yields occurred in years with lower temperatures but with a greater amount of light during the growth stage. In [76], they confirmed that Spanish wheat production showed annual variations of more than 20% depending on climatic conditions.

An important aspect to consider was the reliability of GS. Methodologies used to obtain GS for production and yield varied between NUTS-2. A rigorous methodology for harvest monitoring and prediction is applied in some regions, such as Castile and Leon [77]. Certain outliers were detected in the case of Andalusia and Aragon in GS that affected the performance of models for all VIs. Our results therefore served to confirm results from previous studies, such as [25] and [78], which found this same issue in Spanish GS. Although the article shows that it was possible to estimate wheat yield and production in Spain and its major wheat-producing regions, the differences among regions highlighted how the characteristics of each location and the reliability of official statistics could influence the results. The existence of official yield and production data, as well as data on the precise location of crops, could have an impact on the models in other parts of the world. This could be a potential issue that limits the application of this methodology in regions with a lack of inaccurate data. The low $R^2$ values and high NRE values for the models in Andalusia could be the result of a meteorological change when collecting GS or a problem reporting wheat for the CAP. Two trends were observed that corresponded to two periods: 2006–2009 and 2010–2016 (Figure 12). Data collection was not performed on a national level but by NUTS-2. This way, it was possible that a different methodology was performed in each NUT-2. Moreover, it may have potentially changed over time. European legislation required that EU member countries report quality data on their own statistics. However, Spain had only provided methodological reports, and no metadata or quality reports were available [79]. In fact, EUROSTAT stated how Spain had corrected some overtime breaks

that occurred nationally in 2010, although moderate breaks have continued to occur since 2013 [79]. It was impossible to obtain detailed information about potential errors in data collection for NUTS-2 or at a national level, given this lack of quality reports. Moreover, [25] reported that part of the issue could be the accuracy of Spanish GS, given that the results in Spain were some of the lowest in the European Union according to the MARS Crop Yield Forecasting System [80]. The wheat area shown by GISCAP-CAP in Aragon highlights the possibility of errors arising due to an uneven distribution in adjacent areas of different NUTS-3. It was possible that parcels used for other crops were also included as wheat (Figure 12). This possible error could explain why the area dedicated to wheat according to GISCAP-CAP was far greater than in the GS for this NUTS-2 (Table 3), why the behavior of VIs in Aragon was so different compared to the rest of NUTS- (Figure 4) and why this NUTS-2 was the only one where wheat generally reappeared in the same pixel across several years (Figure 11).

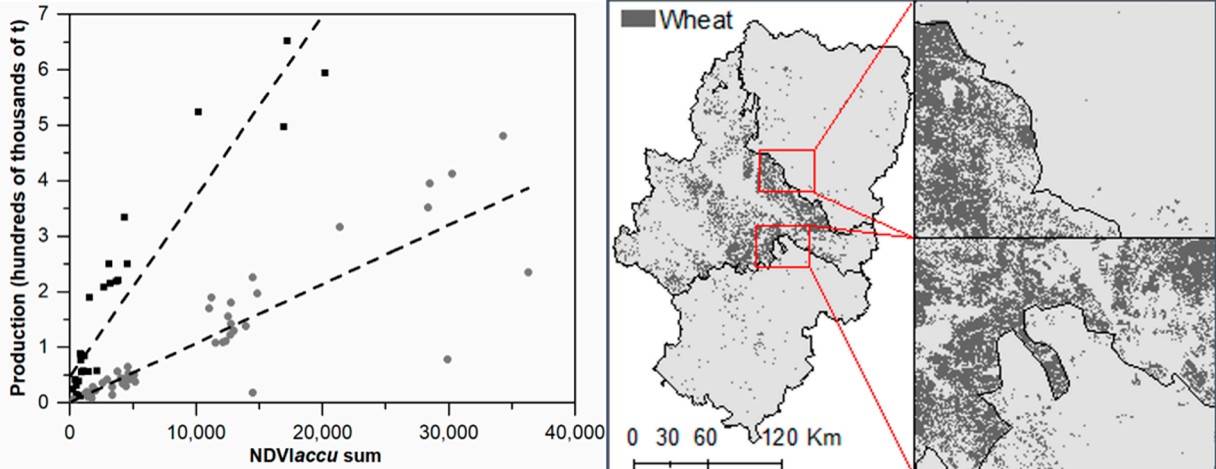

**Figure 12.** Possible data issues that might affect the precision of the models. **Left**: relationship between VIs (e.g., NDVI accumulated summation) and production in Andalusia. Different patterns of behavior can be observed in the 2006–2009 cycle (squares) and the 2010–2016 cycle (circles). **Right**: unequal distribution of wheat fields in adjacent NUTS-3 in Aragon.

The 2006–2009 period for Andalusia and the values for the central NUTS-3 of Aragon were removed to compare how the above-mentioned possible data problems affected the models, and the performance of production models for Spain increased substantially, with $R^2$ values of 0.9 for NDVI, 0.91 for EVI2, and 0.94 for MTCI; and NRE values of 24.11%, 21.72%, and 20.54%, respectively. These values were significant at a 95% confidence level in all cases (Figure 13). Therefore, it can be concluded that this model is precise when using these three Vis to predict wheat production, with MTCI being the VI with the best forecasting capability, in line with other studies [41]. It also highlighted the importance of data analysis to show inaccuracies.

Models that used GISCAP-CAP outperformed those that used CLC (Tables 4–6). The arable land category in the CLC legend includes several crops, not just wheat. There was more aggregation due to its 1 km resolution, which integrates the spectral response of different crops and land covers in each pixel [81,82]. This resolution has been used to distinguish between agricultural or non-agricultural areas [83] or to map natural systems [69], but the high spatial variability and the complexity of agricultural systems required data with a greater spatial resolution. The main reason could be the presence of other crops in the arable land category. The representativeness of these models was also affected by the short length of the time series, with CLC mapping only matching the study period in 2006 and 2012. Therefore, a greater spatial resolution, a more precise wheat mask, and a time window adapted to the phenological conditions of each region allowed a more accurate estimation of wheat yield and production, as proposed by [84] in their study.

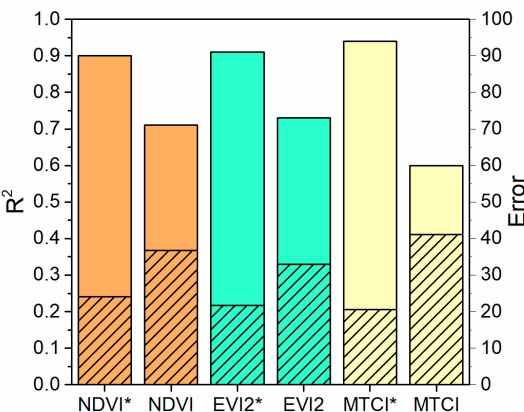

**Figure 13.** $R^2$ and NRE values for production models. NDVI*, EVI2*, and MTCI* do not include Andalusia for the period between 2006 and 2009 nor the central NUTS-3 of Aragon in any year. $R^2$ is represented by colors, and NRE by shading.

## 5. Conclusions

The results showed how the changes proposed in this study with respect to the work of [25] enabled a more accurate estimation of wheat production and yield using VIs in Spain. The selection of custom dates to find the maximum value within each NUTS-2 and the use of wheat-only annual masks from GISCAP-CAP data were key to obtain more accurate models. Production models had a greater forecasting capability than yield models, as $R^2$ values were higher and NRE values were lower. Throughout Spain, production models had $R^2$ values greater than 0.9 for all VIs and NRE values below 25% once some data inaccuracies were removed. With regard to the differences between VIs, MTCI provided the best results, followed by EVI2 and NDVI. Despite NDVI and EVI2 having a better spatial resolution, the fact that MTCI has a higher sensitivity to variations in chlorophyll content may have been a key factor. The results from NDVI and EVI2 were quite similar, but in general, $R^2$ values for EVI2 were slightly higher than NDVI values, and NRE values for EVI2 were slightly lower than NDVI values. The precision of the baseline data had a significant impact on the results. Irregularities in the baseline data caused $R^2$ values to be considerably lower and NRE values to be higher. After omitting values that demonstrated outlier behavior in certain NUTS-2, such as Andalusia and Aragon, the forecasting capability of the models improved significantly. However, there were other factors that had an impact on the models. The median size of the parcels, correlated with the percentage of MODIS pixels actually occupied by wheat fields, also had an impact. For production models, the smaller the size and the lower the percentage, the greater the forecasting capability, while the opposite was true for yield models. Different wheat varieties, differences in climate conditions, and soil properties may have also had an impact. The study has certain limitations, such as the length of the time series, the reliability of GS, and the assumption of the relationship between the period of the maximum VI value and the subsequent month. However, the results obtained show that this methodology can be effectively employed for the optimal estimation of wheat yield and production in Spain. Future studies may benefit from the use of wheat masks created based on remote sensing classification, especially in regions without extensive and reliable data, i.e., GISCAP-CAP. These masks could be created using MODIS or Sentinel-2 data in order to obtain a 10 m spatial resolution.

**Author Contributions:** Conceptualization, M.A.G.-P., V.R.-G., E.S.-R. and V.E.-C.; methodology, M.A.G.-P., V.R.-G., E.S.-R. and V.E.-C.; software, M.A.G.-P.; validation, M.A.G.-P.; formal analysis, M.A.G.-P., V.R.-G. and E.S.-R.; investigation, M.A.G.-P.; resources, M.A.G.-P.; data curation, M.A.G.-P.; writing—original draft preparation, M.A.G.-P.; writing—review and editing, M.A.G.-P., V.R.-G. and E.S.-R.; visualization, M.A.G.-P.; supervision, M.A.G.-P., V.R.-G. and E.S.-R.; project administration, M.A.G.-P.; funding acquisition, M.A.G.-P. and V.R.-G. All authors have read and agreed to the published version of the manuscript.

**Funding:** The first author is an FPU grant holder funded by the Spanish "Ministerio de Universidades" (Reference FPU18/04366). The authors are grateful for the financial support given by the project MIA.2021.M01.004, funded by the Spanish "Ministerio de Asuntos Económicos y Transformación Digital".

**Data Availability Statement:** Common Agrarian Policy data that support the findings of this study are available in "Fondo Español de Garantía Agraria" (FEGA) under request. Restrictions apply to the availability of these data, which were used under license for this study. The property of these data belongs to FEGA and the agriculture departments of Andalusia, Aragon, Castile and Leon, Catalonia, and Extremadura. It makes reference to information that belongs to farmers, so we do not have permission to share these data. GISCAP data that support the findings of this study are openly available in FEGA at https://www.fega.gob.es/en (accessed on 27 September 2023). GS data that support the findings of this study are openly available in Ministerio de Agricultura, Pesca y Alimentación at https://www.mapa.gob.es/es/estadistica/temas/publicaciones/anuario-de-estadistica/default.aspx (accessed on 27 September 2023). MERIS data that support the findings of this study are openly available in ESA EO at https://doi.org/10.5270/EN1-vqoj1gs (accessed on 27 September 2023). MODIS data that support the findings of this study are openly available in USGS EarthExplorer at https://doi.org/10.5067/MODIS/MOD09Q1.006 (accessed on 27 September 2023). CLC data that support the findings of this study are openly available at https://land.copernicus.eu/pan-european/corine-land-cover (accessed on 27 September 2023). Data that support the findings of this study are openly available at https://doi.org/10.17632/8gxrpt4n95.1 (accessed on 27 September 2023).

**Conflicts of Interest:** The authors declare no conflict of interest. The funders had no role in the design of the study; in the collection, analyses, or interpretation of data; in the writing of the manuscript; or in the decision to publish the results.

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
