# Peer review of "Yield Estimation of Wheat Using Cropland Masks from European Common Agrarian Policy: Comparing the Performance of EVI2, NDVI, and MTCI in Spanish NUTS-2 Regions"

_remotesensing, doi:10.3390/rs15225423_

Round 1

Reviewer 1 Report

Comments and Suggestions for Authors

Major Revision:

The study that focuses on monitoring wheat yield and production in Spain using remote sensing data, specifically the Normalized Difference Vegetation Index (NDVI), Enhanced Vegetation Index 2 (EVI2), and MERIS Terrestrial Chlorophyll Index (MTCI). This study demonstrates the effectiveness of using remote sensing data, particularly NDVI, EVI2, and MTCI, to monitor and predict wheat yield and production in Spain. It highlights the significance of accurate wheat masks, the superior performance of CAP data, and the potential for enhancing food security through remote sensing-based monitoring. Additionally, it emphasizes the role of data quality in achieving accurate results.

The details comments are as below:

Major Comments:

1.     The study relies on data from multiple sources, including CAP, GISCAP, GS, and various remote sensing datasets. The limitations and accuracy of these data sources may significantly impact the study's results. The manuscript should discuss the potential errors and uncertainties associated with these datasets.

2.     The manuscript mentions that some data sources, such as CLC masks, were only available for specific years (2006 and 2012). This data sparsity can limit the reliability of the models, especially if important changes occurred in the intervening years.

3.     The manuscript highlights significant differences in the behavior of VIs between regions. The discussion should address the challenges and implications of modeling in regions with diverse climate, agricultural practices, and geographical features.

4.     The study's short time series, spanning only 11 years, may not capture long-term trends and climate variability that can impact wheat production. A more extended dataset could provide more robust insights.

5.     The manuscript suggests that different regions or NUTS-2 may have employed varying data collection methodologies. These differences in methods can introduce bias or inaccuracies, and the manuscript should discuss how these were addressed or acknowledged in the analysis.

6.     The manuscript states that the methodology was replicated from a previous study [25], which identified possible improvements. It would be beneficial to discuss the implications of the proposed improvements, such as the use of crop masks and increased spatial resolution, and how they affect the study's findings.

7.     While the manuscript discusses the performance of models in different regions, it's essential to evaluate the practical implications of these performance differences. Are the models accurate enough for real-world agricultural predictions and management?

8.     The study's assumptions and limitations should be explicitly stated in the manuscript. This includes assumptions about the relationship between VIs and wheat production and the reliability of GS data.

9.     The manuscript mentions the significance of models but does not provide detailed statistical analysis. Providing more in-depth statistical assessments and discussion of statistical methods would enhance the manuscript.

Comments on the Quality of English Language

The quality of the English language in the manuscript to be quite good overall. It's clear and generally well-structured. However, there are a few minor grammatical and stylistic areas where improvements could be made:

Long Sentences: Some sentences are quite lengthy, which can make the text more challenging to follow. Consider breaking them into smaller sentences for improved clarity and readability.

Punctuation: There are instances where more commas could be added to improve sentence structure. For example

Reviewer 2 Report

Comments and Suggestions for Authors

The manuscript aims to find correlations between remote sensing derived vegetation indices and wheat production and yield in Spain. A commonly used methodology is used; the main advances (compared to a paper from 2014), according to the authors, are that they used a higher quality wheat field mask and a more flexible time period to find the maxima of the vegetation indices. 

In my opinion, the manuscript reads well, the methodology is simple but correct, but the novelty is limited. I recommend major revisions and suggest that the authors also check recent literature of similar research on other crops to check the novelty.

I attach a copy of the manuscript with my comments. Some overall comments:

1. Both the introduction and methodology sections need to be improved. There are a number of abbreviations that are not defined and/or not clarified. This manuscript is submitted to Remote Sensing so readers may not be familiar with NUTS-2 regions, GISCAP etc.

2. In the introduction it would be useful to describe why this methodology was chosen.

3. The methodology section is incomplete and needs further description or clarification.

4. In figure 10, top row, it looks to me that the high production data from Catalonia are strongly influencing the correlation. Without these few (ca 9 points) the correlation may not be as good. Please comment on this.

5. The reliability of the GS is mentioned on page 16. This is important. It would be useful to know more about how the GS are calculated; if it is possible to estimate an uncertainty; and the differences across the regions.

Reviewer 3 Report

Comments and Suggestions for Authors

Ensuring global food security relies heavily on wheat production and yields. In Spanish NUTS-2 regions, the efficacy of NDVI, EVI2, and MTCI in evaluating wheat yield was compared using cropland masks. Despite feasibility, the study's methodology requires clearer presentation, and the paper's structure in general requires improvement.

1)   The authors explain the differences in behavior of the VIs in Section 3.1 by utilizing altitude, as seen in Lines 271-283. Thus, it is recommended that a DEM be incorporated into Figure 1 to depict the elevation information.

2)   Section 2 lacks clarity in its methods as they mainly rely on verbal explanations and do not offer specific formulas. To provide greater detail, the authors should provide formulas for NDVI, EVI2, and MTCI; In lines 155-156, when the text mentions "subsequent composites", it remains unclear what exactly it refers to. Additionally, the formula for accumulation must be included; Lines 171-176, it is necessary to display the VI time series and DFT smoothed curve graphically to demonstrate the effectiveness of DFT filtering; Line 210, "We have followed the approach used by previous studies such as:[1] and [57]", the specific methods and formulas used should be detailed in the manuscript.

3)   Lines 334-336: The forecasting ability of production (or yield) models heavily relies on the wheat mask. Hence, it is essential to extract 1-2 years' worth of wheat pixels using remote sensing imagery and compare them with the wheat mask that depends on CLC or CAP data, at the very least.

4)   Superscript and subscript errors (e.g., R2 in section 3.3).

5)   Perhaps colour distinguishes points better than shape in Figure 10-11.

6)   It is recommended that the authors remove the discussion from section 3 and address it separately.

Round 2

Reviewer 1 Report

Comments and Suggestions for Authors

none

Comments on the Quality of English Language

none